# Reproductive Pattern of Parous Women and the Risk of Cancer in Later Life

**DOI:** 10.3390/cancers13153731

**Published:** 2021-07-24

**Authors:** Zahra Pasdar, Neil W. Scott, Lisa Iversen, Philip C. Hannaford, Phyo Kyaw Myint, Sohinee Bhattacharya

**Affiliations:** 1School of Medicine, Medical Sciences and Nutrition, University of Aberdeen, Aberdeen AB25 2ZD, UK; z.pasdar.17@abdn.ac.uk; 2Medical Statistics Group, Institute of Applied Health Sciences, University of Aberdeen, Aberdeen AB25 2ZD, UK; n.w.scott@abdn.ac.uk; 3Academic Primary Care, Institute of Applied Health Sciences, University of Aberdeen, Aberdeen AB25 2ZD, UK; l.iversen@abdn.ac.uk (L.I.); p.hannaford@abdn.ac.uk (P.C.H.); 4Ageing Clinical & Experimental Research Team, Institute of Applied Health Sciences, University of Aberdeen, Aberdeen AB25 2ZD, UK; phyo.myint@abdn.ac.uk; 5Aberdeen Centre for Women’s Health Research, Institute of Applied Health Sciences, University of Aberdeen, Aberdeen AB25 2ZL, UK

**Keywords:** pregnancy, reproductive factors, cancer

## Abstract

**Simple Summary:**

There is evidence to suggest that reproductive factors play an important role in women’s future cancer risk, particularly hormone-dependent malignancies such as breast cancer. However, some controversy remains regarding their effect on any cancer risk. This study aimed to assess the risk of any cancer in a nested case-control study of fertile women living in northeast Scotland in relation to: total number of pregnancies, cumulative time pregnant, age at first pregnancy and interpregnancy interval. Women who were older at first delivery had an increased risk of breast and gastrointestinal cancer, and a reduced risk of cancers of the respiratory tract and uterine cervix, including carcinoma in situ. These findings warrant the investigation of mechanisms underpinning these associations.

**Abstract:**

We assessed the risk of any and site-specific cancers in a case-control study of parous women living in northeast Scotland in relation to: total number of pregnancies, cumulative time pregnant, age at first delivery and interpregnancy interval. We analysed 6430 women with cancer and 6430 age-matched controls. After adjustment for confounders, women with increasing number of pregnancies had similar odds of cancer diagnosis as women with only one pregnancy. The adjusted odds of cancer diagnosis were no higher in women with cumulative pregnancy time 50–150 weeks compared to those pregnant ≤ 50 weeks. Compared with women who had their first delivery at or before 20 years of age, the adjusted odds ratio (AOR) among those aged 21–25 years was 0.81, 95% CI 0.74, 0.88; 26–30 years AOR 0.77, 95% CI 0.69, 0.86; >30 years AOR 0.63, 95% CI 0.55, 0.73. After adjustment, the odds of having any cancer were higher in women who had an inter-pregnancy interval >3 years compared to those with no subsequent pregnancy (AOR 1.17, 95% CI 1.05, 1.30). Older age at first pregnancy was associated with increased risk of breast and gastrointestinal cancer, and reduced risk of invasive cervical, carcinoma in situ of the cervix and respiratory cancer.

## 1. Introduction

Infertility in women has been linked to hormone-dependent malignancies, such as breast and gynaecological cancer in later life [1]. While several studies have shown full term-pregnancy to be protective, especially in relation to breast cancer [2,3], there is an increasing body of evidence showing an association between reproductive history and the risk of other cancers.

Although pregnancy appears to have a protective effect on the development of breast cancer in later life, delayed childbearing seems to negate this effect. In particular, women who have their first child before the age of 20 have a 50% reduction in risk of breast cancer when compared with women who have never had children [4].

Other pregnancy-related factors may play an important role with regard to breast cancer risk in later life. A full-term pregnancy may exert different effects on the risk of breast cancer over time—a short-term increase in risk—possibly due to increased oestrogen production during pregnancy, followed by a long-term protection resulting from the terminal differentiation of mammary tissue [5,6]. If this hypothesis were true, it follows that the protective effect of pregnancy on breast cancer is exerted only when the pregnancy is carried to term.

There is some controversy in the literature regarding the effect of a pregnancy that has been interrupted before full term on the risk of breast cancer. While a study published in the late 20th century observed an increased risk of breast cancer in women with a history of spontaneous or induced abortion [7], Melbye et al. found a slight increase in the risk of breast cancer following delivery before 32 weeks of gestation [8].

A re-analysis of data from 53 different epidemiological studies, however, came to the conclusion that there are no harms associated with induced abortion with regard to breast cancer risk [9]. Subsequent large-scale population-based studies also found no such increase in risk, although the protective effect of a full-term pregnancy was not seen in these studies either [10,11].

There is a dearth of information regarding the association between reproductive history and other types of cancers, including endocrine cancers. Some evidence of association exists between reproductive history and other cancers, including gastric [12], ovarian [13,14], pancreatic [15] and renal [16] as well as overall maternal cancer mortality and specifically breast cancer mortality [17]. However, none of these studies were able to look at the complete reproductive history of individual women and the effect it has on breast and other cancers.

In this study, we hypothesise that pregnancy has a protective effect on overall risk of subsequent cancer in later life, which acts in a dose-dependent manner with an increasing number of full-term pregnancies offering greater protection. To test the hypothesis, we assessed the risk of any cancer in a nested case-control study of parous women living in northeast Scotland in relation to:Total number of pregnanciesCumulative time pregnantAge at first pregnancyInterpregnancy interval

## 2. Materials and Methods

### 2.1. Ethics

Approvals to carry out the study were obtained from the Aberdeen Maternity and Neonatal Databank (AMND) steering committee and the Privacy Advisory Committee of the Information and Services Division (ISD) of the National Health Service (NHS) Scotland.

### 2.2. Sampling Frame

The AMND records and stores all pregnancy-related information about women living in Aberdeen city and district from 1950 onwards. It now has information on over 250,000 pregnancies occurring in these women. Complete reproductive histories of individual women can be constructed using internal data linkage [18]. Records, including identifiers, of individual women who were born before 1968 (i.e., those older than 40 at the time of the data linkage for this study, the age when the incidence of most types of cancer increases substantially) and who had at least their first full-term pregnancy recorded on the AMND were supplied to the ISD of NHS Scotland. Information on baseline characteristics, including age, social class, smoking habits, height and marital status at the time of the first pregnancy, was extracted from the AMND at the time of the identification of these women. Exposure variables of the total number of pregnancies, plus date and outcome of each pregnancy (live birth, miscarriage, ectopic pregnancy, induced abortion or stillbirth) were extracted as well.

The ISD collect and collate information about all hospital admissions, as well as cancer, birth and death registrations, in Scotland in the Scottish Morbidity Records (SMR). The cohort of women identified from the AMND were linked by ISD in 2008 to the national Cancer Registry (SMR06) and to death notifications supplied by the General Register Office (GRO-S) using probability matching. This linkage process involves using identifiers such as name, date of birth and post code to find the closest match. After the linkage, all identifiers were removed by ISD and we were supplied with a pseudo-anonymised (but containing the AMND unit number) and linked study database. All women who were linked to the Scottish registers were then traced on the Community Health Index (CHI) register in Scotland by the Data Management Team, University of Aberdeen (who have access to both registers) to make sure that all linked women had their complete reproductive histories recorded in the AMND. Any woman who had moved out of Aberdeen before the age of 55 years was excluded from the analysis. Figure 1 shows the process of identifying cases and controls.

### 2.3. Study Design

Using the linked dataset as the sampling frame, we conducted a nested case-control study. Cases were all women in the linked dataset with a cancer diagnosis (identified from SMR06) and/or recorded death from cancer (identified from GRO-S). For each case, we randomly identified one control born within two years of the case’s year of birth. Controls did not have a history of cancer at the time of the case’s diagnosis. We excluded cases who had cancer prior to their first pregnancy or during pregnancy. Moreover, women who left Aberdeen before they were likely to have completed their families were also excluded after tracing on the CHI register. This was because we could not be certain of the completeness of their reproductive histories. After applying all the exclusions, there were 6430 cases of cancer and 6430 controls available for statistical analysis.

### 2.4. Definition of Variables

A case was defined as having a record in the Scottish Cancer Registry (SMR06) and/or having a record of death associated with cancer. The site of cancer is coded according to ICD-9 or -10 classifications in this registry and these are presented in the Appendix A with the broad categories used in this analysis. The exposure variables, such as total number of pregnancies and age at first delivery, were all extracted from the AMND directly. The cumulative time pregnant was calculated by adding together the gestational age at the end of each pregnancy. Interpregnancy interval was calculated by subtracting the date of the second delivery from that of the first one and also subtracting the duration of gestation of the subsequent pregnancy. Potential confounders available from the AMND included the Registrar General’s occupation-based social class (non-manual: classes 1, 2 and 3a and manual: classes 3b to 7) and self-reported smoking habit at the time of the first pregnancy. Height, weight and marital status at the time of first pregnancy were also recorded at the time of the first antenatal visit. As weight was poorly recorded for a large proportion of women or recorded at an advanced gestational age, we adjusted for height, which has been shown to be associated with cancer [19].

### 2.5. Statistical Analysis

Personal (social class, smoking habits, height, weight and marital status at time of first pregnancy) and reproductive characteristics consisted of total number of pregnancies, number of pregnancies of normal duration, cumulative time pregnant (in weeks), age at first delivery and inter-pregnancy interval, which were compared between cases and controls using appropriate descriptive statistics. We included in the multivariable analyses variables that were potential confounders and did not have a high proportion of missing data (see below). The non-manual social class formed the reference group for analysis. Similarly, current and ex-smokers were classified together as having ever smoked, with the “never smoked” group as the reference category. Conditional logistic regression was used to calculate the odds ratio and corresponding 95% confidence interval for the development of any cancer among women with different reproductive characteristics, adjusted for height and age at first delivery (except where age at first delivery was being examined). Subgroup analyses were then conducted according to the site of cancer (breast, uterus, invasive uterine cervix, ovary, carcinoma in situ of cervix, skin, respiratory, gastro-intestinal and all other cancers).

### 2.6. Missing Data

There was a substantial proportion of missing data (from one-third to one-half) in the variables weight, social class and smoking status. A large number of women, mostly those who had their first deliveries in the 1950s, had missing data for smoking. As smoking is likely to be an important confounder for some cancers, an additional subgroup analysis by smoking status was conducted including only those women with known smoking history. For clarity of presentation, only the adjusted odds ratios (AORs) are presented for this subgroup analysis. For the other potential confounders, BMI and social class, the proportion of missing data was too high to consider multiple imputation. We did not conduct any further subgroup analyses for these missing data because where data were complete, as they did not appear to vary by case-control status.

## 3. Results

### 3.1. Baseline Characteristics

Women with cancer were younger at the time of their first delivery than controls (23.9 versus 24.5 years, respectively), and more likely to be current or ex-smokers at the time of their first pregnancy (60.3% versus 44.3%; Table 1).

### 3.2. Reproductive Patterns

Following their first delivery, there were 4798 live births, 84 stillbirths, 646 miscarriages, 687 terminations and 49 ectopic pregnancies in the group of women who subsequently developed cancer. Among controls (women not diagnosed with cancer), there were 4660 live births, 78 stillbirths, 687 miscarriages, 689 terminations and 45 ectopic pregnancies following the first delivery.

Table 2 presents the reproductive patterns by case-control status. After adjusting for height and age at first delivery, there was no clear evidence that women with increasing number of pregnancies had higher odds of cancer diagnosis compared with those who had only one pregnancy—AOR 1.03 (95% CI 0.95, 1.13), 1.03 (0.93, 1.15), 1.10 (0.94, 1.28) and 1.10 (0.89, 1.35) for 2, 3, 4 and 5+ pregnancies, respectively. When only pregnancies of normal duration (>37 weeks gestation) were considered, the adjusted odds of cancer in women with two or more pregnancies were not statistically significantly different than in those who had only one pregnancy of normal duration although the unadjusted odds ratios showed an increased odds in women with 2 or more pregnancies. In terms of cumulative time pregnant, the adjusted odds of cancer diagnosis were no different in women who had been pregnant for 50–150 weeks compared to those who had been pregnant for 50 weeks or less in their reproductive life—AOR (95% CI) of 1.07 (0.97, 1.18), 1.08 (0.96, 1.22) and 1.17 (1.00, 1.38) for 51–100, 100–150 and more than 151 weeks, respectively. Once again, on the unadjusted analysis, cumulative time pregnant for more than 50 weeks was significantly associated with a cancer diagnosis.

The inverse relationship between cancer and age at first delivery persisted following adjustment for maternal height. Compared with women who had their first delivery at or before the age of 20 years, the AOR among those who were aged 21–25 years was 0.81, 95% CI 0.74, 0.88; 26–30 years, AOR 0.77, 95% CI 0.69, 0.86; >30 years, AOR 0.63, 95% CI 0.55, 0.73. After adjusting for height and age at first delivery, the odds of having any cancer remained higher in women who had an inter-pregnancy interval greater than 3 years compared with those with no subsequent pregnancy (AOR 1.17, 95% CI 1.05, 1.30).

Table 3 and Table 4 present the subgroup analyses by cancer site. The descriptive statistics for subgroup analyses by cancer site is detailed in Appendix A of the online supplementary files, respectively. Increased adjusted odds ratios for breast cancer were observed in women who were pregnant for longer periods, as reflected in the total number of pregnancies, cumulative time pregnant and shorter inter-pregnancy intervals (Table 3). Having two pregnancies as opposed to one was associated with an increased risk of breast cancer (AOR 1.25, 95% CI 1.01, 1.56); as was three or more pregnancies of normal duration (AOR 1.27, 95% CI 1.01, 1.60). There appeared to be a dose response relationship between the cumulative time a woman was pregnant and breast cancer with an AOR of 1.26 (95% CI 1.01, 1.58) and 1.43 (95% CI 1.10, 1.86) of developing breast cancer if cumulative time pregnant was 51 to 100 weeks or more than 100 weeks, respectively. After adjusting for height, there appeared to be an association between older age at first delivery and breast cancer diagnosis. The odds of having breast cancer were increased in women who had their first delivery after the age of 20 years, although this only reached statistical significance in the age group 26 to 30 years (AOR 1.44, 95% CI 1.11, 1.86). Women with an interpregnancy interval of less than 3 years between pregnancies had statistically significantly higher odds of developing breast cancer (AOR 1.40, 95% CI 1.12, 1.74) than women with no subsequent pregnancies. Age at first delivery of less than 20 years was associated with both invasive cancer of the uterine cervix and carcinoma in situ.

Of the non-gynaecological sites examined (Table 4), only cancer of the gastrointestinal tract was statistically significantly associated with reproductive variables. Similar to breast cancer, the adjusted odds ratios of gastrointestinal cancer were increased in women who had two pregnancies (regardless of duration) compared to one. Women who were pregnant for 51 to 100 weeks during their reproductive life had a higher chance of developing gastrointestinal cancer compared to those who were pregnant for 50 weeks or less (AOR 1.50, 95% CI 1.11, 2.03). Compared to those who were aged 20 or younger at the time of their first delivery, women who delivered when they were over the age of 31 had a significantly higher risk of developing gastrointestinal cancer in later life. None of the other site-specific cancers showed any clear evidence of associations with the reproductive variables examined apart from interpregnancy interval of greater than 3 years was associated with an increased risk of other cancers.

Table 5 presents the main analysis for any cancer diagnosis, stratified by women whose smoking habits were recorded at the time of their first delivery. Non-smokers who had four pregnancies of normal duration as opposed to one, those who were pregnant for 151 weeks or more during their reproductive life or those who were older than 26 years at the time of their first delivery had significantly lower odds of developing any cancer in later life after adjustment for height and age at delivery. In current or past smokers, age at delivery when younger than 20 years was associated with an increased risk of cancer.

## 4. Discussion

To the best of our knowledge, this is the first study which explored several pregnancy-related factors and their association with any cancer in parous women. Our results showed that the odds of developing any cancer were statistically significantly reduced with increasing age at first delivery. Cancer diagnosis was also associated with inter-pregnancy intervals longer than 3 years. However, breast cancer risk was increased in women with delayed childbearing and with longer time pregnant, as reflected by higher cumulative time pregnant and shorter interpregnancy intervals. Older age at first delivery was associated with a reduced risk of invasive cancer of the uterine cervix, carcinoma in situ and cancers of the respiratory tract.

Although there have been several reports of associations between infertility and hormone-dependent cancer [1,20,21], there is controversy regarding the effect of reproductive patterns in parous women on subsequent cancer risk. It was shown as early as 1926 by Lane-Claypon [22] that increasing parity was a protective factor for breast cancer. Later, MacMahon et al. conducted a multicentre case-control study in seven countries all over the world and found that younger maternal age at first term birth, rather than total parity, protected against breast cancer [23]. Several subsequent studies have confirmed these findings [2,24,25,26]. We found slightly different results. Whilst we also found that breast cancer risk was increased in women who had their first delivery after age 20 years, in a subgroup analysis, being pregnant for longer increased the chances of breast cancer. There could be several explanations for this different pattern of risk. First, the reference category in our study for maternal age at first delivery analysis was less than 20 years, and it is possible that the protective effect of earlier age at first delivery disappeared even earlier than previously thought. The positive association seen with cumulative time pregnant may be explained by the longer duration the breast is subjected to pregnancy hormones. This hypothesis is consistent with the excess risk of breast cancer observed in oral contraceptive users [27]. The protective effect of high parity observed by others may be explained by the duration of breast feeding [28].

Although the reduction of risk seen with cumulative time pregnant (>101 weeks) on cancer of the uterus was not statistically significant in this study, a recent analysis found that whilst there is a greater magnitude of reduction in risk for a full-term pregnancy than an early pregnancy loss, each additional pregnancy is associated with a further reduction in endometrial cancer risk [29].

The similar reproductive pattern seen in our subgroup analysis of women with gastrointestinal cancer has not been described before, although a Swedish national record linkage study involving 406,439 women did not find a protective effect of increasing parity on the development of stomach cancer [12].

The main strength of this study lies in the quality of the data recorded in the databases (AMND and the Scottish Registers) used for record linkage. These databases are almost 100% complete in terms of case ascertainment and have been in existence since the 1950s, thereby reducing the possibility of selection bias. Many other studies have utilised women’s recall of reproductive events, which may be subject to recall bias. We are confident in the completeness of our records of reproductive history, as they are recorded contemporaneously at the time of pregnancy. Moreover, we excluded all women who could not be traced on the CHI register before they were likely to have completed their families. Cases and controls were matched on year of birth. This controlled for an important confounder, as cancer risk varies by age and over time.

An obvious limitation of this study is that we only looked at women who had at least one full-term pregnancy, thereby missing any “all or none” protective effect of pregnancy and term delivery on subsequent development of cancer. Another weakness of this study was the inability to adjust for potential confounding factors, notably breast feeding, BMI, exogenous hormone use and smoking. The AMND contains information about breast feeding at discharge from hospital. We did not adjust for this variable in our analyses for two reasons—first, data were missing in a substantial proportion of pregnancies; second, even where they were available, feeding status at discharge from hospital does not capture the duration of breast feeding, which is of more relevance to breast cancer risk. We were able to do a subgroup analysis where smoking data were available, which achieved similar results overall. The women were identified from the AMND, and although they are representative of the local population, the observed findings may not be generalisable to a wider population. Moreover, we had relatively few women with higher-order parity in our dataset, so our study could have been underpowered to observe the protective effect seen in previous reports. Another limitation is multiple statistical testing and significant findings due to chance alone.

Interestingly, breast cancer risk showed a different direction of effect of reproductive factors, such as interpregnancy interval to other cancers risk. As the other cancer categories contained a mixed bag of different cancers, as shown in the Appendix A, with no specific site predominating, it would be presumptuous of us to draw any conclusions from this. This may possibly be a chance finding as numbers of any particular subtype categorised as “others” was low.

Despite the limitations, this study shows that reproductive patterns in parous women could be used to identify women at risk of developing cancer in later life. Together with genetic and environmental risk factors, such as smoking, reproductive patterns may be included in prediction models to assess the risk of cancer in women at an early age. Preventing teenage pregnancies would have the added benefit of reducing the risk of cancer. Future research needs to focus on the mechanistic explanations of the associations found in this and other epidemiological studies. For example, this cohort predated the rollout of HPV immunisations in Scotland. The association shown in this study between younger age at first delivery with cervical cancer may be explained by increased rates of HPV infection in women who were sexually active at a younger age.

## 5. Conclusions

For women who had been pregnant at least once, there was evidence of a small increase in the risk of any cancer among those having subsequent pregnancies. Older age at first pregnancy was associated with increased risk of breast and gastrointestinal cancer and a reduced risk of invasive cervical, carcinoma in situ of the cervix and respiratory cancer.

## Figures and Tables

**Figure 1 cancers-13-03731-f001:**
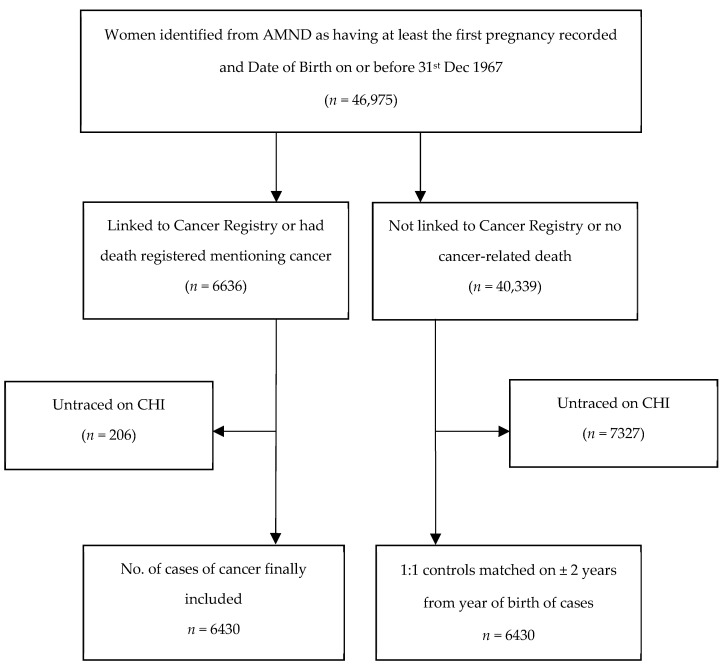
Flow chart of selection of cases (women with cancer) and controls (women without cancer).

**Table 1 cancers-13-03731-t001:** Comparison of baseline characteristics between cases and controls at the time of first pregnancy.

Baseline Characteristics		Cases (Max *n* = 6430)	Controls (Max *n* = 6430]
		Mean (SD) [*n*]	Mean (SD) [*n*]
Age at first delivery		23.9 (4.9) [*n* = 6420]	24.5 (5.0) [*n* = 6422]
Height (cm)		159.8 (6.0) [*n* = 6385]	159.5 (6.1) [*n* = 6384]
BMI		23.3 (3.3) [*n* = 2135]	23.7 (3.6) [*n* =2 309]
		*n* (% of known)	*n* (% of known)
Social class	Manual	3442 (68.9%)	3611 (69.8%)
	Non-Manual	1553 (31.1%)	1559 (30.2%)
	Not known	1435	1260
Ever smoked?	No	1029 (39.7%)	1435 (55.7%)
	Yes	1564 (60.3%)	1142 (44.3%)
	Not known	2577	2593
Marital status	Single	686 (10.7%)	474 (7.4%)
	Married/cohabiting	5703 (88.7%)	5908 (91.9%)
	Divorced/separated	40 (0.6%)	46 (0.7%)
	Not known	1	2

**Table 2 cancers-13-03731-t002:** Comparison of reproductive variables between women with (cases) and without (controls) cancer.

Total Number of Pregnancies	Cases (*n* = 6429) *n* (%)	Controls (*n* = 6429) *n* (%)	OR * *n* = 12,858	Adj OR ** *n* = 12,644
1 (reference)	1620 (25.2%)	1763 (27.4%)	1	1
2	2758 (42.9%)	2775 (43.2%)	1.08 (0.99, 1.18)	1.03 (0.95, 1.13)
3	1320 (20.5%)	1258 (19.6%)	**1.14 (1.03, 1.26)**	1.03 (0.93, 1.15)
4	489 (7.6%)	429 (6.7%)	**1.24 (1.07, 1.44)**	1.10 (0.94, 1.28)
5+	242 (3.8%)	204 (3.2%)	**1.30 (1.06, 1.58)**	1.10 (0.89, 1.35)
Not known	1	1		
No. of pregnancies of normal duration	Cases (*n* = 6429) *n* (%)	Controls (*n* = 6429) *n* (%)	OR * *n* = 12,858	Adj OR ** *n* = 12,644
1 (reference)	1416 (22.0%)	1589 (24.7%)	1	1
2	2451 (38.1%)	2397 (37.3%)	**1.15 (1.05, 1.26)**	1.10 (1.00, 1.21)
3	1356 (21.1%)	1353 (21.0%)	**1.13 (1.02, 1.26)**	1.03 (0.92, 1.15)
4	687 (10.7%)	621 (9.7%)	**1.25 (1.10, 1.43)**	1.11 (0.96, 1.27)
5+	519 (8.1%)	469 (7.3%)	**1.25 (1.08, 1.45)**	1.07 (0.92, 1.25)
Not known	1	1		
Cumulative time pregnant (weeks)	Cases [*n* = 5718] *n* (%)	Controls [*n* = 5757] *n* (%)	OR * *n* = 10,262	Adj OR ** *n* = 10,108
≤50	1469 (25.7%)	1639 (28.5%)	1	1
51–100	2568 (44.9%)	2590 (45.0%)	**1.11 (1.01, 1.22)**	1.07 (0.97, 1.18)
100–150	1169 (20.4%)	1096 (19.0%)	**1.19 (1.06, 1.33)**	1.08 (0.96, 1.22)
151+	512 (9.0%)	432 (7.5%)	**1.34 (1.15, 1.57)**	1.17 (1.00, 1.38)
Not known	712	673		
Age at first delivery	Cases [*n* = 6419] *n* (%)	Controls [*n* = 6421] *n* (%)	OR * *n* = 12,822	Adj OR *** *n* = 12,644
≤20	1705 (26.6%)	1439 (22.4%)	1	1
21–25	2656 (41.4%)	2731 (42.5%)	**0.81 (0.74, 0.89)**	**0.81 (0.74, 0.88)**
26–30	1417 (22.1%)	1477 (23.0%)	**0.79 (0.71, 0.88)**	**0.77 (0.69, 0.86)**
31+	641 (10.0%)	774 (12.1%)	**0.65 (0.57, 0.75)**	**0.63 (0.55, 0.73)**
Not known	11	9		
Interpregnancy interval	Cases (*n* = 5895) *n* (%)	Controls (*n* = 5904) *n* (%)	OR * *n* = 10,832	Adj OR ** *n* = 10,694
No subsequent pregnancy	1662 (28.2%)	1823 (30.9%)	1	1
<3 years	2606 (44.2%)	2611 (44.2%)	**1.11 (1.02, 1.22)**	1.05 (0.96, 1.16)
≥3 years	1627 (27.6%)	1470 (24.9%)	**1.23 (1.12, 1.37)**	**1.17 (1.05, 1.30)**
Not known	535	526		

* Conditional logistic regression. ** Conditional logistic regression adjusted for height and age at first delivery. *** Conditional logistic regression adjusted for height. Statistically significant odds ratios are shown as bold.

**Table 3 cancers-13-03731-t003:** Subgroup analyses by cancer site (breast/gynaecological cancers): reproductive variables by case-control status.

Reproductive Pattern	Breast	Uterus	Uterine Cervix	Ovary	Carcinoma In Situ
	Adj OR	Adj OR	Adj OR	Adj OR	Adj OR
Total number of pregnancies	*n* = 2386	*n* = 228	*n* = 296	*n* = 202	*n* = 3080
1 (reference)	1	1	1	1	1
2	**1.25 (1.01, 1.56)**	0.89 (0.46, 1.70)	1.12 (0.61, 2.04)	1.82 (0.83, 3.98)	0.91 (0.74, 1.13)
3+	1.21 (0.96, 1.52)	0.96 (0.46, 2.00)	0.95 (0.50, 1.80)	0.66 (0.31, 1.44)	1.01 (0.81, 1.26)
No. of pregnancies of normal duration	*n* = 2386	*n* = 228	*n* = 296	*n* = 202	*n* = 3080
1 (reference)	1	1	1	1	1
2	1.16 (0.95, 1.42)	0.86 (0.45, 1.64)	1.04 (0.60, 1.81)	1.68 (0.81, 3.52)	**0.81 (0.67, 0.98)**
3+	**1.27 (1.01, 1.60)**	0.91 (0.47, 1.78)	1.37 (0.73, 2.56)	0.62 (0.27, 1.43)	0.85 (0.69, 1.05)
Cumulative time pregnant (weeks)	*n* = 1900	*n* = 186	*n* = 222	*n* = 156	*n* = 2532
≤50	1	1	1	1	1
51–100	**1.26 (1.01, 1.58)**	0.91 (0.47, 1.78)	1.01 (0.54, 1.88)	1.16 (0.50, 2.68)	0.90 (0.73, 1.11)
101+	**1.43 (1.10, 1.86)**	0.90 (0.38, 2.14)	1.36 (0.64, 2.86)	0.46 (0.17, 1.24)	0.92 (0.72, 1.17)
Age at first delivery *	*n* = 2386	*n* = 228	*n* = 296	*n* = 202	*n* = 3080
≤20	1	1	1	1	1
21–25	1.24 (1.00, 1.55)	1.19 (0.62, 2.29)	**0.23 (0.12, 0.46)**	1.36 (0.58, 3.18)	**0.64 (0.54, 0.76)**
26–30	**1.44 (1.11, 1.86)**	2.15 (0.91, 5.11)	**0.37 (0.16, 0.86)**	1.12 (0.48, 2.61)	**0.39 (0.32, 0.49)**
31+	1.36 (0.97, 1.90)	0.69 (0.18, 2.70)	0.12 (0.03, 1.05)	0.52 (0.15, 1.79)	**0.22 (0.16, 0.31)**
Interpregnancy interval	*n* = 2038	*n* = 204	*n* = 246	*n* = 172	*n* = 2394
No subsequent pregnancy	1	1	1	1	1
<3 years	**1.40 (1.12, 1.74)**	0.77 (0.39, 1.51)	1.27 (0.71, 2.27)	1.14 (0.52, 2.54)	0.85 (0.69, 1.05)
>3 years	1.22 (0.95, 1.57)	1.11 (0.51, 2.45)	1.60 (0.82, 3.15)	1.42 (0.61, 3.31)	0.94 (0.74, 1.19)

Conditional logistic regression adjusted for height and age at first delivery. * Conditional logistic regression adjusted for height. N (Ca): number of cases, N (Co): number of controls (NB: not all are included in the conditional logistic regression because of missing data in covariates). Statistically significant odds ratios shown in bold.

**Table 4 cancers-13-03731-t004:** Subgroup analyses by cancer site (non-gynaecological cancers): reproductive variables by case-control status.

Reproductive Pattern	Skin	Respiratory	Gastrointestinal	Other Cancers
	Adj OR	Adj OR	Adj OR	Adj OR
No of pregnancies of normal duration	*n* = 684	*n* = 942	*n* = 1312	*n* = 3146
1 (reference)	1	1	1	1
2	0.92 (0.60, 1.43)	1.35 (0.95, 1.92)	**1.35 (1.01, 1.79)**	1.21 (1.00, 1.46)
3+	0.89 (0.57, 1.37)	1.27 (0.88, 1.84)	1.21 (0.90, 1.63)	1.00 (0.82, 1.22)
Total no. of pregnancies	*n* = 684	*n* = 942	*n* = 1312	*n* = 3146
1 (reference)	1	1	1	1
2	1.09 (0.72, 1.65)	1.29 (0.93, 1.80)	**1.37 (1.04, 1.81)**	1.14 (0.95, 1.36)
3+	0.99 (0.64, 1.53)	1.33 (0.92, 1.92)	1.28 (0.95, 1.72)	1.05 (0.86, 1.28)
Cumulative time pregnant (weeks)	*n* = 574	*n* = 692	*n* = 1070	*n* = 2498
≤50	1	1	1	1
51–100	0.89 (0.57, 1.40)	1.19 (0.82, 1.73)	**1.50 (1.11, 2.03)**	1.17 (0.96, 1.43)
101+	0.90 (0.55, 1.45)	1.29 (0.84, 1.97)	1.27 (0.91, 1.78)	1.15 (0.91, 1.44)
Age at first delivery *	*n* = 684	*n* = 942	*n* = 1312	*n* = 3146
≤20	1	1	1	1
21–25	0.99 (0.66, 1.48)	0.70 (0.49, 1.00)	1.22 (0.88, 1.70)	0.83 (0.69, 1.00)
26–30	0.80 (0.50, 1.26)	**0.61 (0.40, 0.93)**	1.43 (0.97, 2.09)	0.99 (0.79, 1.25)
31+	0.64 (0.35, 1.16)	**0.54 (0.31, 0.95)**	**1.73 (1.08, 2.76)**	0.94 (0.71, 1.26)
Interpregnancy interval	*n* = 584	*n* = 808	*n* = 1190	*n* = 2736
No subsequent pregnancy	1	1	1	1
<3 years	0.87 (0.57, 1.33)	1.09 (0.77, 1.55)	1.27 (0.96, 1.68)	1.08 (0.89, 1.30)
>3 years	0.81 (0.50, 1.30)	1.35 (0.93, 1.95)	1.31 (0.97, 1.78)	**1.33 (1.08, 1.63)**

Conditional logistic regression adjusted for height and age at first delivery. * Conditional logistic regression adjusted for height. N (Ca): number of cases, N (Co): number of controls (NB: not all are included in the conditional logistic regression because of missing data in covariates). Statistically significant odds ratios are shown in bold.

**Table 5 cancers-13-03731-t005:** Subgroup analyses including only those with known smoking status: reproductive variables by case-control status.

Reproductive Pattern	Non-Smoker			Current/Past Smoker		
	N (Ca)	N (Co)	Adj OR	N (Ca)	N (Co)	Adj OR
No. of pregnancies of normal duration			*n* = 2439			*n* = 2664
1 (reference)	379	252	1	252	309	1
2	581	427	1.00 (0.81, 1.23)	478	614	0.94 (0.76, 1.16)
3	302	212	0.88 (0.69, 1.13)	248	357	0.99 (0.78, 1.27)
4	117	91	0.92 (0.66, 1.28)	100	159	1.03 (0.75, 1.42)
5+	56	47	0.93 (0.60, 1.44)	64	125	1.19 (0.82, 1.72)
Total	1435	1029		1142	1564	
Total number of pregnancies			*n* = 2439			*n* = 2664
1 (reference)	416	282	1	301	395	1
2	695	489	0.92 (0.75, 1.12)	559	730	0.88 (0.73, 1.07)
3	243	212	1.03 (0.80, 1.33)	206	296	0.91 (0.71, 1.16)
4	64	37	0.63 (0.40, 0.99)	59	107	1.07 (0.74, 1.55)
5+	17	9	0.55 (0.24, 1.27)	17	36	1.19 (0.65, 2.18)
Total	1435	1029		1142	1564	
Cumulative time pregnant (weeks)			*n* = 2251			*n* = 2390
≤50	389	260	1	268	336	1
51–100	664	458	0.93 (0.76, 1.15)	523	676	0.92 (0.75, 1.13)
100–150	225	179	0.99 (0.76, 1.29)	188	277	0.97 (0.74, 1.26)
151+	62	33	0.62 (0.39, 0.98)	53	105	1.22 (0.83, 1.79)
Total	1340	930		1032	1394	
Age at first delivery *			*n* = 2439			*n* = 2664
≤ 20	256	240	1	354	628	1
21–25	486	378	0.83 (0.66, 1.03)	446	565	0.72 (0.60, 0.86)
26–30	398	269	0.70 (0.55, 0.90)	220	265	0.67 (0.53, 0.84)
31+	292	139	0.50 (0.38, 0.65)	121	106	0.48 (0.35, 0.64)
Total	1432	1026		1141	1564	
Interpregnancy interval			*n* = 2204			*n* = 2355
No subsequent pregnancy	425	290	1	296	373	1
<3 years	573	419	0.96 (0.79, 1.18)	434	592	0.94 (0.77, 1.16)
>3 years	294	220	0.97 (0.77, 1.23)	277	415	1.02 (0.82, 1.28)
Total	1292	929		1007	1380	

Logistic regression adjusted for height and age at first delivery; * Logistic regression adjusted for height; N (Ca): number of cases, N (Co): number of controls.

## Data Availability

The data presented in this study are available on request from the corresponding author after relevant permissions. The data are not publicly available to ensure that the proposed use of the linked AMND-SMR data conforms to accepted scientific standards in terms of methodology, confidentiality and ethics.

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
