# Peer review of "Reproductive Pattern of Parous Women and the Risk of Cancer in Later Life"

_cancers, 2021, doi:10.3390/cancers13153731_

Round 1

Reviewer 1 Report

The authors have tried to correlate the reproductive patters of parous women with cancer risk. The result are interested and there are some minor errors

  1. Title of 3.3 section should be statistical analysis and the title that is there should be moved to the bottom of the table.
  2. Some of the tables present a lot of information and is hard to understand it, try to transform some of the data in the tables into graphs.

Reviewer 2 Report

The manuscript by Pasdar et al. is an extensive analysis of the association between number of pregnancies, total time pregnant, inter-pregnancy intervals, and maternal age at first pregnancy with the development of any subsequent cancer. Many previous studies have limited this type of analysis to breast cancer development exclusively without including the development of other types of cancers. This is an important study that advances the field. That being said, many of their findings for development of breast cancer are at odds with their findings on the development of any cancers. For instance, on line 112, the authors find that women with cancer were younger at the time of first delivery; however, on line 144 they found that an older age at first delivery was more associated with development of breast cancer and on line 159 found that GI cancer risk increased with the age of delivery (especially over 31 years old). Furthermore, they state on line 131 that an interpregnancy interval greater than or equal to 3 years was associated with increased odds of developing any cancer, while on line 147 they associated an interpregnancy interval less than 3 years with increased odds of developing breast cancer. Additionally, they found on line 168 that within non-smokers, either 4 pregnancies instead of one or spending greater than or equal to 151 weeks pregnant lowered the odds of getting any cancer while on line 141 found that greater than or equal to 2 pregnancies instead of 1 was associated with an increased risk of breast cancer and on line 156 found that increased pregnancies and/or a longer time spent pregnant increased the gastrointestinal cancer risk. If possible, it would be helpful to know which types of cancers are driving the phenotypes observed in the “any cancers” categories, especially since they seem to be opposite to the specific development of breast cancer, one of if not the most common cancers among women. A smaller aside, there were findings about increased risk of cervical cancers in mothers less than 20 years old at first delivery; including a comment on HPV or, if possible, rates of HPV divided by age would be helpful. Finally, mentioning in the introduction or methods the reasoning behind adjusting for height (association between height and cancer) would also be helpful.

Author Response

We would like to thank reviewer 2 for their insightful comments on the manuscript. We have tried to take the suggestions on board as far as possible and have revised the manuscript accordingly. Please see responses below in green.

  1. Many of their findings for development of breast cancer are at odds with their findings on the development of any cancers. If possible, it would be helpful to know which types of cancers are driving the phenotypes observed in the “any cancers” categories, especially since they seem to be opposite to the specific development of breast cancer, one of if not the most common cancers among women. Many thanks for pointing this out. We have now added a few sentences to clarify this varying direction of effect in the discussion section as follows: 

    Interestingly breast cancer risk showed a different direction of effect of reproductive factors such as interpregnancy interval to other cancers risk. As the other cancers category contained a mixed bag of different cancers as shown in the supplementary table with no specific site predominating, it would be presumptuous of us to draw any conclusions from this. This may possibly be a chance finding as numbers of any particular subtype categorized as “others” was low.

  2. A smaller aside, there were findings about increased risk of cervical cancers in mothers less than 20 years old at first delivery; including a comment on HPV or, if possible, rates of HPV divided by age would be helpful. We have added a couple of sentences regarding HPV in the discussion section as follows: For example, this cohort predated the rollout of HPV immunisations in Scotland. The association shown in this study between younger age at first delivery with cervical cancer may be explained by increased rates of HPV infection in women who were sexually active at a younger age.
  3. Finally, mentioning in the introduction or methods the reasoning behind adjusting for height (association between height and cancer) would also be helpful. We thank the reviewer for this suggestion and we have added a sentence in the methods section along with a reference justifying the adjustment for height as follows: 

    As weight was poorly recorded for a large proportion of women or recorded at an advanced gestational age, we adjusted for height which has been shown to be associated with cancer. (Nunney L. 2018 Size matters: height, cell number and a person’s risk of cancer. Proc. R. Soc. B 285: 20181743. http://dx.doi.org/10.1098/rspb.2018.1743)